# Local Administration of Low-Dose Nerve Growth Factor Antibody Reduced Pain in a Rat Osteoarthritis Model

**DOI:** 10.3390/ijms22052552

**Published:** 2021-03-04

**Authors:** Yuan Tian, Tomohiro Onodera, Mohamad Alaa Terkawi, Koji Iwasaki, Ryosuke Hishimura, Dawei Liang, Takuji Miyazaki, Norimasa Iwasaki

**Affiliations:** 1Department of Orthopedic Surgery, Faculty of Medicine and Graduate School of Medicine, Hokkaido University, Kita-15, Nish-7, Kita-ku, Sapporo 060-8638, Japan; tyfree1001@gmail.com (Y.T.); materkawi@med.hokudai.ac.jp (M.A.T.); rockcape324@gmail.com (K.I.); 527079901ldw@gmail.com (D.L.); takuzimiyazaki@gmail.com (T.M.); niwasaki@med.hokudai.ac.jp (N.I.); 2Global Institution for Collaborative Research and Education (GI-CoRE), Hokkaido University, Sapporo 060-0808, Japan; 3Department of Orthopedic Surgery, Hokkaido University Hospital, Kita14, Nishi5, Kita-Ku, Sapporo 060-8648, Japan; hishi_piero@yahoo.co.jp

**Keywords:** osteoarthritis, pain, nerve growth factor (NGF), intra-articular injection

## Abstract

Systemic injection of a nerve growth factor (NGF) antibody has been proven to have a significant relevance in relieving osteoarthritis (OA) pain, while its adverse effects remain a safety concern for patients. A local low-dose injection is thought to minimize adverse effects. In this study, OA was induced in an 8-week-old male Sprague–Dawley (SD) rat joint by monoiodoacetate (MIA) injection for 2 weeks, and the effect of weekly injections of low-dose (1, 10, and 100 µg) NGF antibody or saline (control) was evaluated. Behavioral tests were performed, and at the end of week 6, all rats were sacrificed and their knee joints were collected for macroscopic and histological evaluations. Results showed that 100 µg NGF antibody injection relieved pain in OA rats, as evidenced from improved weight-bearing performance but not allodynia. In contrast, no significant differences were observed in macroscopic and histological scores between rats from different groups, demonstrating that intra-articular treatment does not worsen OA progression. These results suggest that local administration yielded a low effective NGF antibody dose that may serve as an alternative approach to systemic injection for the treatment of patients with OA.

## 1. Introduction

Osteoarthritis (OA) is the most common type of arthritis that affects more than 300 million people globally and contributes to an economic burden on both patients and society [1,2]. According to the recent Osteoarthritis Research Society International white paper, OA has been considered a serious disease because of the lack of any efficient treatment [3]. Clinically, the symptoms of OA include pain, joint stiffness, and disability that lead to a decline in patients’ quality of life, the loss of social labor, and an economic burden on the whole society. Pain is particularly important in all clinical problems, as it is not only the cause of hospital visits for treatment but also the main reason underlying poor quality of life and social labor loss [4]. Current pharmacological treatment for OA pain using traditional analgesics, such as non-steroidal anti-inflammatory drugs and acetaminophen, is partly effective and accompanied with serious side-effects, such as disruption of the gastrointestinal mucosa ulceration, cardiovascular toxicity, and suppression of platelet aggregation [5,6]. Therefore, more effective treatments that relieve OA pain are warranted.

A humanized immunoglobulin G2 monoclonal nerve growth factor (NGF) antibody that has been used as an analgesic agent for OA has recently gained significant relevance in relieving OA-associated pain in a clinical trial [7]. Intravenous injections could effectively improve chronic pain and joint function in patients with OA at a dose of 5 or 10 mg every 8 weeks as compared with a placebo [8]. However, a clinical phase III study of tanezumab (NGF antibody) was held by the Food and Drug Administration in 2015 because of its adverse effect, as all patients presented with progressively worsening OA and subsequently required total joint replacement in one of 13 phase III studies [7]. Moreover, other adverse effects, such as paresthesia, arthralgia, pain in the extremities, and headaches were also observed after systemic administration of tanezumab, and these effects remain a safety concern for patients [9,10,11].

Considering the high effectiveness of NGF antibody treatment for relieving chronic pain, such as OA pain, researchers continue to focus on developing NGF antibody treatment. As OA only affects a limited number of joints, intra-articular injection therapy appears to be a more attractive alternative for patients than other treatments [12]. Local injection can largely decrease the risk of systemic exposure and the incidence of adverse effects. Moreover, local injection is thought to reduce the effective dosage, possibly preventing the aggravation of adverse effects and decreasing the economic burden on patients [13,14]. Although local administration of the NGF antibody, such as its intra-articular injection, might be a preferable way to maintain its effectiveness for the treatment of chronic pain and to reduce the incidence of adverse effects, the analgesic effects of local treatment with a low-dose NGF antibody on OA pain and related adverse effects on cartilage degeneration have not yet been investigated. The purpose of this study was to investigate the effect of low-dose intra-articular injections of the NGF antibody on OA joints using a rat model.

## 2. Results

### 2.1. The NGF Antibody Can Relieve the Pain and Improve the Weight-Bearing Performance but Not Allodynia

To observe the effect of analgesic local treatment on OA, a murine model of monoiodoacetate (MIA)-induced OA was employed [15]. To confirm the effect of local treatment with the NGF antibody, different doses (1, 10, and 100 µg) of the NGF antibody were intra-articularly injected into the right knees of rats once a week from the end of week 2. Behavioral tests were performed twice a week to observe the pain behavior in animals. The results of the behavioral tests showed that MIA injection impaired the weight-bearing performance (Appendix A) and decreased the threshold, termed as allodynia (Appendix A), as a sign of pain from the first week. Rats receiving saline injection did not show any pain behavioral changes. The intra-articular injection of 100 µg of the NGF antibody effectively relieved the pain in the OA model rat, as evidenced from improved weight-bearing performance (Figure 1a, saline, 1 µg, 10 µg, vs. 100 µg from week 3, Appendix A), whereas 1 and 10 µg doses showed no pain-relieving effects. However, allodynia, which was also induced by MIA injection, did not improve after NGF antibody injection at any concentrations tested (Figure 1b, Appendix A). These results show that only 100 µg of NGF antibody could relieve the OA pain induced by MIA injection, as confirmed from the improvement in weight-bearing asymmetry but not allodynia.

### 2.2. The NGF Antibody Injection Exerts No Negative Effect on the Cartilage

To observe the effect of the NGF antibody on the joint pathological progress, macroscopic and histological scores were evaluated. The results of macroscopic evaluations showed that MIA injection damaged the cartilage, imitating OA characterized with cartilage erosions (Figure 2a). No erosion was reported in sham knee joint cartilage treated with saline. The injection of the NGF antibody at all doses and saline had no evident adverse effects on the joints (Figure 2b). Consistent with these results, histological evaluations were performed based on hematoxylin and eosin (H&E; Figure 3a) and safranin O staining (Figure 3b). MIA inhibited the function of glyceraldehyde-3-phosphatase-induced cell death, resulting in disorganized cartilage structure, reduction in safranin-O staining, and destruction of tidemark integrity [16]. These pathological changes that appeared at the end of week 6 indicated the damage to the rat knee joint cartilage. H&E (Figure 3a) and safranin O staining (Figure 3b) showed no significant difference in each MIA group, indicating that NGF antibody injection exhibited no negative effects on cartilage pathology (Figure 3c). However, during the progression of MIA-induced OA, NGF antibody injection did not obviously interrupt the pathological progression of OA.

Fluorescence staining for the NGF revealed the MIA injection-mediated increase in the concentration of NGF in the synovial tissue surrounding the affected joints (Figure 4). Injection of the NGF antibody neutralized the NGF in the tissue and, consequently, reduced the signal of NGF staining.

## 3. Discussion

NGF plays an important role in pain and can serve as a signal in inflammatory joint disease [17,18]. NGF antibody treatment has been proven to be effective to relieve chronic pain, such as OA pain. However, high-dose administration of the NGF antibody (5 mg/kg) is accompanied with adverse effects, including progressively worsening cartilage degeneration and potential nerve system side-effects, such as paresthesia, arthralgia, and headaches, and is considered problematic in OA [10,19]. Hochberg revealed the rapid progression in OA after systemic treatment with the NGF antibody and concluded that it is imperative to use low effective doses to control the risk of adverse effects [20]. Moreover, Bélanger et al. reviewed the evaluation of safety data of systemic treatment with the NGF antibody and found safety concerns both in clinical and nonclinical cases [21]. Therefore, a low effective dose should be considered to decrease the risk of side-effects and systemic exposure.

In this study, we found that a low dose of 100 μg NGF antibody could alleviate MIA-induced pain, suggesting that intra-articular administration of a low dose of NGF antibody may be an effective treatment for OA, specifically when a limited number of arthritic joints are effected, such as mono- or oligo-articular OA. Several studies have focused on the efficacy and safety of the NGF antibody for chronic pain using an animal OA model through systemic treatment. In general, systemic injection necessitates a 10 mg/kg dose, corresponding to 5 mg/injection [22,23]. Aso et al. found that the inhibition of tropomyosin receptor kinase (TrkA), a high-affinity receptor of the NGF, reduced pain in meniscal transection-operated rats after oral treatment with 30 mg/kg twice daily [24]. Additionally, systemic administration of the NGF antibody in rat models increased limbs edema [25,26]. In fact, intra-articular injection has been considered a more cost-effective treatment for OA than systemic injection [10,27]. Direct delivery of drugs necessitates a low yet effective dose, which can decrease the risk of side-effects and damage to other unimpaired tissues. Therefore, local treatment with the NGF antibody may be an appropriate strategy to reduce the dosage and thereby the incidence of side-effects and exposure of the whole body to the antibody. This is the first study to elaborate on the pain-relieving effect of local treatment with the NGF antibody and its effects on articular cartilage.

Systemic injection is known to improve both weight-bearing asymmetry and mechanical allodynia [22]. The intra-articular injection of the NGF antibody could only improve weight-bearing performance. Mechanical allodynia is a painful sensation stimulated by light touch. Although the mechanism of allodynia is incompletely understood, it is thought to involve alterations in mechano-transduction and sensory neurons of the central nervous system (CNS) [28,29]. NGF can bind to two receptors, a high-affinity TrkA receptor and a low-affinity p75 neurotrophic receptor (p75NTR). Upregulated levels of NGF and TrkA have been reported in the synovial fluid of some patients with arthritis [18,30]. The binding of NGF to TrkA results in the retrograde transport of the resulting complex to the cell body of sensory neurons located in the dorsal root ganglia (DRG) [31]. The increased expression of NGF under OA conditions contributes to changes in receptor sensitivity or DRG, which may result in central sensitization [32,33].

The limitation of this study is the insufficient analgesic effect on allodynia-related pathologies, such as complex regional pain syndrome after intra-articular injection of the NGF antibody, as suggested from our results. Whether or not the local injection of the NGF antibody can effectively reverse the CNS changes established during the NGF rising phase requires further exploration and experiments. Furthermore, although OA pain was relieved by NGF antibody injection, complications of OA, such as cartilage degeneration, still require treatment. Patients treated with the NGF antibody may have better ability for physical activities because of the absence of pain in the OA joint, thereby aggravating cartilage damage and eventually leading to early joint replacement. Moreover, the mechanism by which topical administration of the NGF antibody does not suppress allodynia without exacerbating OA is still unknown. Further studies are warranted to clarify the association between these two phenomena.

In conclusion, the intra-articular administration of a low dose of NGF antibody could reduce pain but not allodynia or, more importantly, cartilage degeneration in rat. Although doses and intervals need to be considered in humans due to a species mismatch, the local administration may serve as a safe alternative approach to systemic injection for pain relief treatment in OA patients.

## 4. Materials and Methods

### 4.1. Ethics Statement

All animal experiments were approved on 20 February 2018 by the Institute of Animal Care and Use Committee of the Hokkaido University Graduate School of Medicine (no. 17–0136).

### 4.2. MIA-Induced Rat OA Pain Model

Eight-week-old male Sprague–Dawley (SD) rats (CLEA Tokyo, Japan) were housed on a 12 h light/dark cycle and had free access to food and water. Rats were randomly divided into six groups, including four MIA injection groups and two sham groups (saline injection). Each group contained a sample size of six rats. Rats were anesthetized by intraperitoneal injections of 100 and 10 mg/kg ketamine and xylazine, respectively, before local injection. Later, 0.5 mg MIA (Nacalai Tesque, Kyoto, Japan) dissolved in 25 µL saline solution was injected once into the right knee joint capsules of rats from MIA groups through the infrapatellar ligament to induce OA-like pain [34]. The same volume of saline solution was injected into the right knee joints of sham rats. To confirm the effect of MIA and saline local injection, behavioral tests, such as weight-bearing and von Frey filament tests, were continuously performed before and after local injection for 6 weeks.

### 4.3. Behavioral Tests

To define the pain behavior performance of rat knee joints, direct pain behavior performance (for detecting weight-bearing asymmetry) and indirect pain behavior performance (for detecting allodynia) were tested in this study. Both behavioral evaluations were performed twice a week for each rat from the week before local injection. All data were collected and applied for statistical analysis after 6 weeks, and data were combined to show weekly changes in behavior tests.

Weight-bearing changes in the hind paw represented the weight distribution between the right (operated) and left (control) limbs as a direct index of joint pain in the osteoarthritic knee [35]. A static weight-bearing test device (Bioseb, Chaville, France) was used to determine the hind paw weight distribution. Rats were rested in an angled chamber so that each hind paw was placed on a separate force testing plate. The force exerted by each hind limb (measured in grams) was averaged over a 5-s period. Each data point was the mean of the three tests of 5-s reading. Results were presented as the distribution of weight-bearing between the left (control) limb and right (operated) limb, as calculated by the following equation: bearing weight of operated leg/bearing weight of both legs × 100%. Decreasing the distribution of weight-bearing can be considered a direct index of joint pain.

A von Frey filament (Shin factory, Fukuoka, Japan) was used to measure the mechanical threshold for indicating allodynia, which was induced by mechanical stimulation. Rats were placed in a chamber with a mesh bottom, which allowed access to the plantar surface of each hind paw. The animals were allowed to acclimatize in the chamber for 10 min before testing. The mechanical threshold of the ipsilateral hind paw was assessed using the modified up-down method [36]. A von Frey hair was perpendicularly applied to the plantar surface of the ipsilateral hind paw until the hair flexed and held in place for 3 s. The von Frey hair (range, 0.4–60.0 g) was applied in ascending order to observe the withdrawal reaction of rats. If a rapid withdrawal reaction was observed, the von Frey hair was subsequently applied in descending order until the withdrawal reaction was no longer observed. This method was repeated thrice at an interval of 10 min. Allodynia was confirmed from the decrease in the mechanical threshold as compared to the original mechanical threshold before injection of MIA or saline.

### 4.4. Treatment with the NGF Antibody

To observe the effect of the local administration of the NGF antibody on OA pain, saline and different doses (1, 10, and 100 µg) of the NGF antibody (Mochida Pharmaceutical Co., Ltd., Tokyo, Japan) were injected into the right knee joint capsules of rats from MIA groups through the infrapatellar ligament at week 2. The injection was administered once a week, four times until the end of week 6 (Figure 5). The left knee joints remained non-operated and were considered control groups.

### 4.5. Cartilage Degradation Evaluation

Rats were sacrificed at the end of week 6. The knee joints were collected and fixed in 10% formalin and subjected to cartilage degradation evaluations, including macroscopic and histological scoring, in a blinded manner.

Macroscopic scores were assessed after joint collection and evaluated using the Likert scale (Table 1) [37].

Regarding the performance of knee joint histological examination, knee joints were decalcified in decalcifying solution B (Wako, Osaka, Japan) for 3 days before the joints were embedded in paraffin. Sections of 5 µm were prepared and subjected to H&E and safranin O staining. The degeneration of the cartilage was microscopically examined using an all-in-one microscope (Keyence, Osaka, Japan) and scored using the modified Mankin scoring system (Table 2) [38].

### 4.6. Immunofluorescence Staining

To observe the inhibition of NGF in the knee joint capsule, 5-μm sections of knee joints were prepared and blocked with horse serum. The sections were subsequently incubated for overnight at 4 °C with a primary NGF antibody (1:500, Mochida Pharmaceutical Co. Ltd., Tokyo, Japan). The primary antibody was detected with a treatment with goat anti-mouse Alexa Fluor Plus 594 (Invitrogen, Waltham, MA, USA) for 60 min at 37 °C. The cell nuclei were stained with 4′,6-diamidino-2-phenylindole (DAPI, Invitrogen, Carlsbad, CA, USA). Finally, the sections were mounted, covered with cover slides, and examined using a fluorescence microscope (Keyence, Osaka, Japan).

### 4.7. Statistical Analyses

For behavioral evaluation, a two-way analysis of variance (ANOVA), followed by Tukey’s test, was used for the comparison of each group. For macroscopic and histological evaluations, one-way ANOVA, followed by Tukey’s multiple-comparison procedure, was used to compare differences among groups (GraphPad Software, La Jolla, CA, USA). Results presented as mean ± standard error of the mean (SEM) were considered statistically significant at *p* < 0.05.

## 5. Conclusions

Based on the results of this study, the effective dose of the NGF antibody was significantly lower with intra-articular injection than that with systemic injection without acceleration in OA progression. Moreover, intra-articular injection might be an alternative approach to systemic injection for the treatment of patients with OA. Further experiments are necessary to improve the intra-articular injection treatment with the NGF antibody to treat cartilage degeneration and elucidate the mechanism of allodynia in OA, as well as the relationship between the NGF antibody and allodynia.

## Figures and Tables

**Figure 1 ijms-22-02552-f001:**
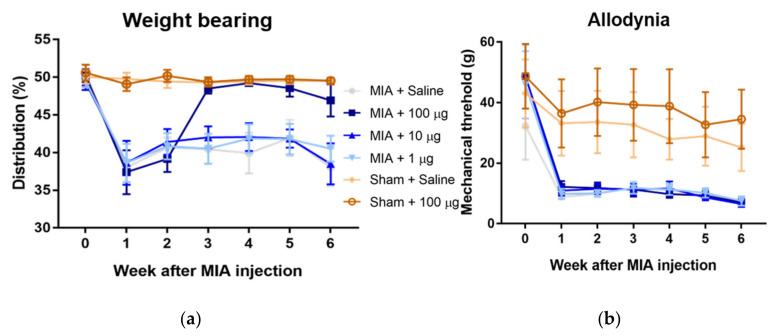
Anti-nerve growth factor (NGF) antibody treatment relieved the pain in a rat osteoarthritis (OA) model, as evidenced by improved weight-bearing performance but not allodynia. (**a**) Monoiodoacetate (MIA) injection can induce weight-bearing asymmetry, whereas a saline injection did not show any significant changes in the rat’s weight-bearing performance. Moreover, 100 μg of anti-NGF antibody treatment reduced pain in the rat OA model, as evident from the improvement in weight-bearing performance (*p* < 0.0001). Furthermore, 1 and 10 μg antibody treatment did not improve the weight-bearing performance after MIA injection. The results represent the 95% confidence intervals for six rats. (**b**) MIA injection significantly lowered rat hind paw withdrawal mechanical thresholds compared to saline injection. No significant improvement in allodynia was observed after the injection of the NGF antibody at any doses. The results represent the 95% confidence intervals for six rats.

**Figure 2 ijms-22-02552-f002:**
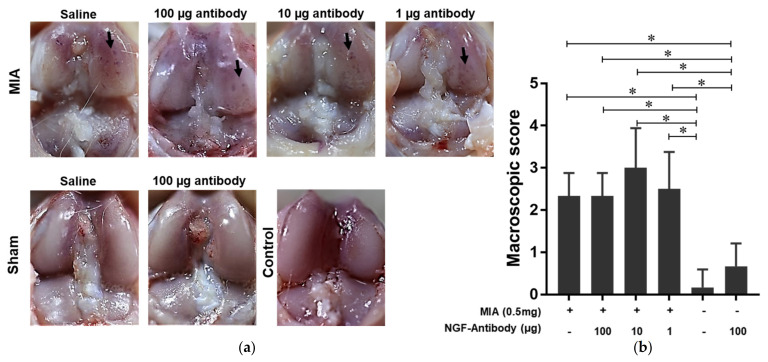
Macroscopic evaluation of rat-affected knee joints indicates no differences among the MIA injection groups. (**a**) The macroscopic figures showed that MIA injection can induce cartilage degradation, whereas saline injection had no effect. Arrows indicate cartilage erosions. (**b**) Macroscopic score using the Likert scale showed that the nerve growth factor antibody injection had no evident effect. The results represent 95% confidence intervals for six rats. * indicates a significant difference, as determined by one-way analysis of variance, followed by the Tukey’s multiple-comparison procedure (*p* ≤ 0.05).

**Figure 3 ijms-22-02552-f003:**
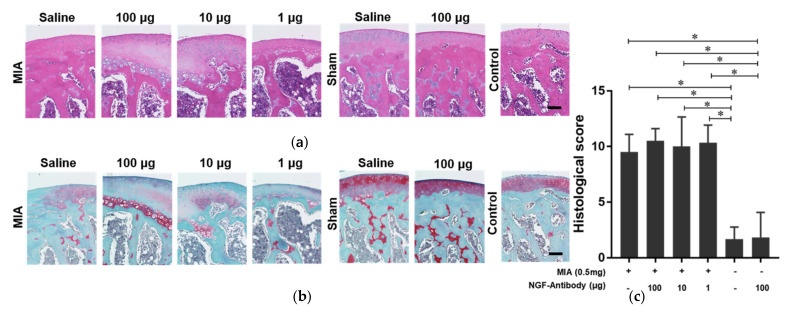
Histological evaluation of rat-affected knee joints consistent with macroscopic evaluation. (**a**) Hematoxylin and eosin (H&E) staining showed that the inflammation formed around the cartilage and chondrocytes was disorganized and no longer observed after MIA injection. The scale bar is 100 μm. (**b**) Results of safranin O staining showed that MIA injection induced cartilage degradation, characterized with cartilage irregularities and reduction in staining intensity. The scale bar is 100 μm. (**c**) The results of the Mankin score showed no significant difference among the MIA groups or between the sham groups, revealing that the treatment of the anti-NGF antibody did not exacerbate the pathological progression of OA joints. The results represent the 95% confidence intervals for six rats. * indicates a significant difference, as determined by one-way analysis of variance, followed by the Tukey’s multiple-comparison procedure (*p* ≤ 0.05).

**Figure 4 ijms-22-02552-f004:**
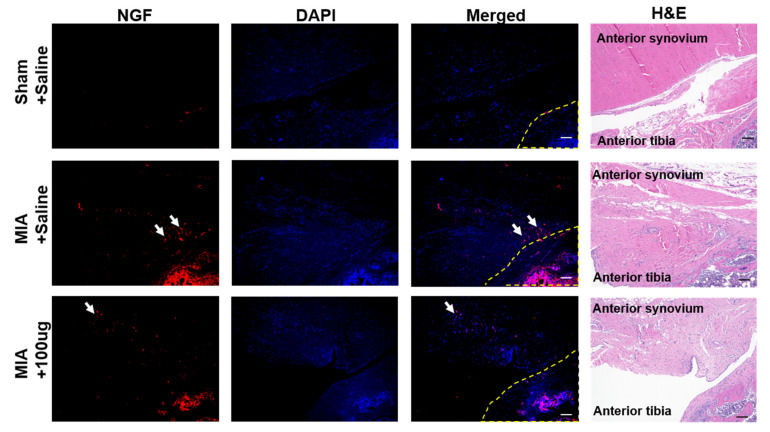
Fluorescent staining of knee joints indicates NGF concentration changes before and after treatment. MIA injection induced a NGF increase, which appears as a high NGF positive area. After the injection of 100 μg NGF, there was a decrease of NGF positive reaction. Arrows indicate the positive reaction for the NGF. The yellow dotted line indicates the area of the articular cartilage and subchondral bone tissue. DAPI; 4′,6-diamidino-2-phenylindole. The scale bar is 100 μm.

**Figure 5 ijms-22-02552-f005:**
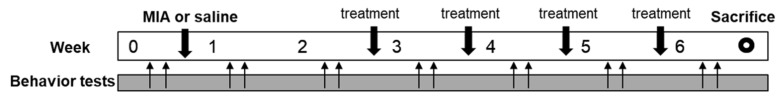
Timeline of local injection of the NGF antibody in an MIA-induced OA rat model.

**Table 1 ijms-22-02552-t001:** Likert scale (Guingamp macroscopic lesions).

Grade
0 = normal appearance
1 = slight yellowish discoloration of the chondral surface
2 = little cartilage erosion in load-bearing areas
3 = large erosions extending down to the subchondral bone
4 = large erosions with large areas of subchondral bone exposure.

**Table 2 ijms-22-02552-t002:** Modified Mankin scoring system.

Cartilage Structure	
Normal	0
Surface irregularities	1
Pannus and surface irregularities	2
Clefts to transitional zone	3
Clefts to radial zone	4
Clefts to calcified zone	5
Complete disorganization	6
Cartilage cells	
Normal	0
Pyknosis, lipid degeneration hypercellularity	1
Clusters	2
Hypocellularity	3
Safranin-O	
Normal	0
Slight reduction	1
Moderate reduction	2
Severe reduction	3
No staining	4
Tidemark integrity	
Intact	0
Destroyed	1

## Data Availability

The data presented in this study are available in results section, figures and Appendix A.

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
