# Peer review of "Local Administration of Low-Dose Nerve Growth Factor Antibody Reduced Pain in a Rat Osteoarthritis Model"

_ijms, 2021, doi:10.3390/ijms22052552_

Round 1
Reviewer 1 Report
Tian et al. used the MIA-induced OA model to analyze the effect of low-dose anti-NGF treatment. The problem is that even though anti-NGF antibodies do reduce pain in OA patients, this treatment exerts adverse side effects. In patients treated with high doses of anti-NGF antibodies OA was progressively worsening eventually leading to joint replacement. Therefore, the hypothesis was that low doses might reduce pain but at the same time not affect cartilage degeneration.
In general, the topic is interesting and it might have clinical relevance and could even change clinical practice. However, there are a couple of points that the authors need to address:
It is rather difficult to compare the doses applied in the murine model and in human clinical trials, respectively. Can the concentrations be normalized somehow so that a comparison is possible? Based on the data obtained, what doses would the authors now recommend for the treatment of OA patients? Can human and murine system be compared at all? Could it make sense to show that also in the murine system (and the same setting as described here) cartilage degeneration would be worsened with high doses?
The authors should make sure that they mention all relevant studies and trials that used anti-NGF treatment in their introduction.
It is unusual to present the experimental setup in the last figure. I would either recommend showing it in the Methods section or as Figure 1.
I would recommend to present Figure S1 also in the main text.
The arrows in Figure 2a are not easy to see. The sections in Figure 3a and 3b (100µg) seem to be from a different area or in a different plane than the other sections. The cellular density is much higher and I am uncertain if a comparison with the other groups is reasonable.
Line 140: the reference to figure 4 makes no sense.
Line 156: the statements about mono- or oligom-articualt OA needs a better explanation
Line 190: NGF antibodies do not really prevent OA exacerbation…they might reduce pain but not allodynia and more importantly also not cartilage degeneration…please revise this statement
Line 240: Treatment ‚with‘ insteadt of ‚of‘
The authors should carefully check if all abbreviations were introduced and used systematically.
Author Response
Comments and Suggestions for Authors
Tian et al. used the MIA-induced OA model to analyze the effect of low-dose anti-NGF treatment. The problem is that even though anti-NGF antibodies do reduce pain in OA patients, this treatment exerts adverse side effects. In patients treated with high doses of anti-NGF antibodies OA was progressively worsening eventually leading to joint replacement. Therefore, the hypothesis was that low doses might reduce pain but at the same time not affect cartilage degeneration.
In general, the topic is interesting and it might have clinical relevance and could even change clinical practice. However, there are a couple of points that the authors need to address:
- It is rather difficult to compare the doses applied in the murine model and in human clinical trials, respectively. Can the concentrations be normalized somehow so that a comparison is possible? Based on the data obtained, what doses would the authors now recommend for the treatment of OA patients? Can human and murine system be compared at all? Could it make sense to show that also in the murine system (and the same setting as described here) cartilage degeneration would be worsened with high doses?
We acknowledge the reviewer comment and know that it’s difficult to compare the doses applied in the murine model and in human clinical trials. In clinical trial, the frequency of adverse effect increased with the increase of NGF antibody dose (Alan J Kivitz, et al. 2013. Pain.), which indicate that it’s necessary to use the lowest effective doses to alleviate risk. Also, there are researches showed that the systemic injection of NGF antibody increased hindpaw edema in the limb of rats, while low dose systemic administration at dose 0.1mg/kg showed no deterioration of rats’ limbs edema (Ilya Sabsovich, et al. 2008. Pain; G. Ishikawa, et al. 2015. OAC).
However, it was impossible to standardize the dose of human and mouse strictly as reviewer commented. It is considered that it is valuable in the human clinic that low dose NGF antibody local administration can obtain the analgesic effect without the exacerbation of cartilage degeneration which exists in the murine model.
We added the description regarding to species differences and its limitation in the discussion section as below:
Although doses and intervals need to be considered in humans due to the species mismatch, the local administration may serve as a safe alternative approach to systemic injection for pain relief treatment in OA patients. Line 202-204.
- The authors should make sure that they mention all relevant studies and trials that used anti-NGF treatment in their introduction.
We acknowledge the reviewer comment and we have adjusted the reference to mention more relevant studies. Line 47, 53, 168. Reference 8, 11, 25, 26.
Reference 8, in the clinical trial, Thomas J Schnitzer, et al found out that subjects received partial symptomatic relief of OA pain with NSAIDs may receive greater benefit with tanezumab monotherapy.
Reference 11, Thomas J. Schnitzer et al described patients with moderate to severe OA of the knee or hip and inadequate response to standard analgesics, tanezumab, compared with placebo, resulted in statistically significant improvements in scores assessing pain and physical function, although the improvements were modest and tanezumab-treated patients had more joint safety events and total joint replacements.
Reference 25, A single dose of anti-NGF antibody exerts a long-lasting analgesic effect on pain during motion in a rat model of OA but accompanied by a statistically significant increase in knee edema compared to vehicle control.
Reference 26, the studies of show that anti-NGF administered after tibia fracture exerts a strong anti-nociceptive effect, a modest sparing effect on bone loss and no ability to reduce vascular abnormalities or cytokine production.
- It is unusual to present the experimental setup in the last figure. I would either recommend showing it in the Methods section or as Figure 1.
We have modified the experimental setup place as suggested referred as Figure 5. Line 257.
- I would recommend to present Figure S1 also in the main text.
We acknowledge the reviewer comment and we have moved Figure S1 to the main text referred as Figure 4. Line 138.
- The arrows in Figure 2a are not easy to see. The sections in Figure 3a and 3b (100µg) seem to be from a different area or in a different plane than the other sections. The cellular density is much higher and I am uncertain if a comparison with the other groups is reasonable.
We acknowledge the reviewer comment and we have modified Figure 2a and changed the 100 µg section figure in Figure 3a and 3b.
- Line 140: the reference to figure 4 makes no sense.
We acknowledge the reviewer comment and have remove the reference. Line 140.
- Line 156: the statements about mono- or oligom-articualt OA needs a better explanation
We acknowledge the reviewer comment and we added the explanation as below:
, specifically when limited number of arthritic joints are effected, such as mono- or oligo-articular OA.
Line 161-162.
- Line 190: NGF antibodies do not really prevent OA exacerbation…they might reduce pain but not allodynia and more importantly also not cartilage degeneration…please revise this statement
Thank you very much for your valuable comment. Comments from the Reviewer are one of the facts we want to describe the most. We mentioned this in a conclusion paragraph separate from limitations section:
In conclusion, the intra-articular administration of a low dose of NGF antibody could reduce pain but not allodynia and more importantly also not cartilage degeneration. Although doses and intervals need to be considered in humans due to the species mismatch, the local administration may serve as a safe alternative approach to systemic injection for pain relief treatment in OA patients.
Line 200 - 204.
- Line 240: Treatment ‚with‘ insteadt of ‚of‘
We acknowledge the reviewer comment and modified in the text. Line 251.
- The authors should carefully check if all abbreviations were introduced and used systematically.
We acknowledge the reviewer comment and we have modified the abbreviations. Line 16, 87-88, 113, 123, 125.
Reviewer 2 Report
I have peer reviewed the manuscript titled: “Local administration of low-dose nerve growth factor antibody reduced pain in a rat osteoarthritis model”. The purpose of this experimental study was to investigate the effect of low-dose intraarticular injections of a nerve growth factor (NGF) antibody on osteoarthritis (OA) joints. The topic of this manuscript is of importance for treatment of osteoarthritis pain, because of safety concerns of systemic treatment with NGF antibody including worsening cartilage degeneration and potential other side-effects such as paresthesia, arthralgia, and headache. Generally, the manuscript has been well written and I find it interesting. However, I have some concerns on the paper.
- Line 47: please add references at the end of this sentence.
- Please unify the units: in the manuscript, for NGF doses, mg was used in one place (line 154) and µg in another (line 20, 73, 81)
- In my opinion, the statement provided several times in the text by the authors (line 190, 279) that NGF antibodies given intraarticularly prevents OA exacerbation is too far-reaching conclusion because results of this study showed that intraarticular NGF antibody injection relieved only pain in OA rats, but not allodynia and no significant differences were observed in macroscopic and histological examination between groups of rats. Intraarticular NGF antibody injection exhibited no negative effects on cartilage pathology, but also did not interrupt the pathological progression of OA. In order to determine whether NGF antibody administered intraarticularly has a preventive effect, NGF should be administered before the experimental induction of osteoarthritis, i.e. intraarticular administration of MIA. Please reconsider this part of conclusions and change in the text.
Author Response
I have peer reviewed the manuscript titled: “Local administration of low-dose nerve growth factor antibody reduced pain in a rat osteoarthritis model”. The purpose of this experimental study was to investigate the effect of low-dose intraarticular injections of a nerve growth factor (NGF) antibody on osteoarthritis (OA) joints. The topic of this manuscript is of importance for treatment of osteoarthritis pain, because of safety concerns of systemic treatment with NGF antibody including worsening cartilage degeneration and potential other side-effects such as paresthesia, arthralgia, and headache. Generally, the manuscript has been well written, and I find it interesting. However, I have some concerns on the paper.
- Line 47: please add references at the end of this sentence.
We acknowledge the reviewer comment and we have added the references [8] at the end of that sentence. Line 47.
- Please unify the units: in the manuscript, for NGF doses, mg was used in one place (line 154) and µg in another (line 20, 73, 81)
We acknowledge the reviewer comment and unify the units as recommended. Line 159.
- In my opinion, the statement provided several times in the text by the authors (line 190, 279) that NGF antibodies given intraarticularly prevents OA exacerbation is too far-reaching conclusion because results of this study showed that intraarticular NGF antibody injection relieved only pain in OA rats, but not allodynia and no significant differences were observed in macroscopic and histological examination between groups of rats. Intraarticular NGF antibody injection exhibited no negative effects on cartilage pathology, but also did not interrupt the pathological progression of OA. In order to determine whether NGF antibody administered intraarticularly has a preventive effect, NGF should be administered before the experimental induction of osteoarthritis, i.e. intraarticular administration of MIA. Please reconsider this part of conclusions and change in the text.
We totally agreed with the reviewer comment and modified the expression of our conclusions. Line 201, 300, 304.
Round 2
Reviewer 2 Report
I have read the authors' responses to my remarks. As for point 1, the authors cited the work Schnitzer et al.(2015), in which patients received intravenous tanezumab (5 or 10 mg) or placebo every 8 week with or without naproxen or celecoxib, and efficacy was assessed as change from baseline to week 16. The sentence from the manuscript is: “Single intravenous injections could effectively improve chronic pain and joint function in patients with OA at a dose of 5 mg every 16 8 weeks as compared with placebo [8]”. Thus there are some inconsistencies in the text of the manuscript. Additionally, in my opinion, we either write about single drug administration or about administration every 8 weeks. So please correct it. Otherwise, I have no more objections, after making this correction I accept the manuscript.
Author Response
Comments and Suggestions for Authors
I have read the authors' responses to my remarks. As for point 1, the authors cited the work Schnitzer et al.(2015), in which patients received intravenous tanezumab (5 or 10 mg) or placebo every 8 week with or without naproxen or celecoxib, and efficacy was assessed as change from baseline to week 16. The sentence from the manuscript is: “Single intravenous injections could effectively improve chronic pain and joint function in patients with OA at a dose of 5 mg every 16 8 weeks as compared with placebo [8]”. Thus there are some inconsistencies in the text of the manuscript. Additionally, in my opinion, we either write about single drug administration or about administration every 8 weeks. So please correct it. Otherwise, I have no more objections, after making this correction I accept the manuscript.
We acknowledge the reviewer comment and we have corrected the description regarding intravenous injections of tanezumab in human as below (Line 45-47);
Intravenous injections could effectively improve chronic pain and joint function in patients with OA at a dose of 5 or 10 mg every 8 weeks as compared with placebo [8].